Host-Microbe Biology

# Depletion of *Blautia* Species in the Microbiota of Obese Children Relates to Intestinal Inflammation and Metabolic Phenotype Worsening

Alfonso Benítez-Páez,[a] Eva M. Gómez del Pugar,[a] Inmaculada López-Almela,[a] Ángela Moya-Pérez,[a] Pilar Codoñer-Franch,[b,c] Yolanda Sanz[a]

[a]Microbial Ecology, Nutrition, and Health Research Unit, Institute of Agrochemistry and Food Technology, Spanish National Research Council (IATA-CSIC), Paterna-Valencia, Spain
[b]Department of Pediatrics, Obstetrics and Gynecology, University of Valencia, Valencia, Spain
[c]Department of Pediatrics, Dr. Peset University Hospital, Valencia, Spain

**ABSTRACT** Cross-sectional studies conducted with obese and control subjects have suggested associations between gut microbiota alterations and obesity, but the links with specific disease phenotypes and proofs of causality are still scarce. The present study aimed to profile the gut microbiota of lean and obese children with and without insulin resistance to characterize associations with specific obesity-related complications and understand the role played in metabolic inflammation. Through massive sequencing of 16S rRNA gene amplicons and data analysis using a novel permutation approach, we have detected decreased incidence of *Blautia* species, especially *Blautia luti* and *B. wexlerae*, in the gut microbiota of obese children, which was even more pronounced in cases with both obesity and insulin resistance. There was also a parallel increase in proinflammatory cytokines and chemokines (gamma interferon [IFN-γ], tumor necrosis factor alpha [TNF-α], and monocyte chemoattractant protein 1 [MCP-1]) in feces of obese children compared to those of lean ones. *B. luti* and *B. wexlerae* were also shown to exert an anti-inflammatory effect in peripheral blood mononuclear cell cultures *in vitro*, compared to non-obesity-associated species. We suggest that the depletion of *B. luti* and *B. wexlerae* species in the gut ecosystem may occur in cases of obesity and contribute to metabolic inflammation leading to insulin resistance.

**IMPORTANCE** Child obesity constitutes a risk factor for developing insulin resistance which, if sustained, could lead to more severe conditions like type 2 diabetes (T2D) in adulthood. Our study identified previously unknown species whose depletion (*Blautia luti* and *Blautia wexlerae*) is associated with insulin resistance in obese individuals. Our results also indicate that these bacterial species might help to reduce inflammation causally linked to obesity-related complications. Childhood is considered a window of opportunity to tackle obesity. These new findings provide, therefore, valuable information for the future design of microbiota-based strategies for the early prevention of obesity-related complications.

**KEYWORDS** *Blautia luti*, *Blautia wexlerae*, childhood obesity, insulin resistance, gut microbiota, inflammation, permubiome, PBMCs, children, gut inflammation, microbiota, obesity, probiotics

This article followed an open peer review process. The review history can be read here.

Address correspondence to Alfonso Benítez-Páez, abenitez@iata.csic.es, or Yolanda Sanz, yolsanz@iata.csic.es.

Blautia species could alleviate intestinal inflammation underlying child obesity

Obesity represents a major concern for public health because of its rising prevalence and associated comorbidity (1–3). Childhood obesity is of special concern, as the number of overweight children and adolescents has increased 10-fold in the last 40 years (4). Of these individuals, over 60% are expected to remain overweigh in early

**TABLE 1** Anthropometry and serum biochemistry of children included in this study

| Parameter analyzed[a] | Value for child group | | | P value of statistical analysis[b] | | |
|---|---|---|---|---|---|---|
| | Lean (n = 16) | Obese (n = 20) | Obese+IR (n = 15) | Lean vs obese | Lean vs obese+IR | Obese vs obese+IR |
| Age (yrs) | 10.06 ± 0.80 | 11.30 ± 0.63 | 13.27 ± 0.66 | 0.616 | **0.010** | 0.152 |
| Sex[c] | Boys, 12; girls, 4 | Boys, 10; girls, 10 | Boys, 7; girls, 8 | 0.236 | 0.329 | 1.000 |
| BMI (z-score) | 1.68 ± 0.08 | 2.42 ± 0.17 | 2.76 ± 0.15 | **0.001** | **<0.001** | 0.140 |
| HOMA-IR[d] | 1.37 ± 0.13 | 2.28 ± 0.15 | 5.18 ± 0.55 | **0.002** | **<0.001** | **<0.001** |
| Glucose (mg/dl) | 89.67 ± 1.63 | 92.15 ± 1.34 | 87.31 ± 3.97 | 0.245 | 0.588 | 0.208 |
| Insulin (mg/dl) | 6.06 ± 0.54 | 10.38 ± 0.70 | 25.63 ± 2.04 | **<0.001** | **<0.001** | **<0.001** |
| Total cholesterol (mg/dl) | 156.70 ± 6.79 | 145.50 ± 3.82 | 160.60 ± 6.05 | 0.139 | 0.669 | **0.042** |
| HDL cholesterol (mg/dl) | 50.00 ± 2.54 | 32.24 ± 2.33 | 37.36 ± 3.36 | **<0.001** | **0.005** | 0.207 |
| LDL cholesterol (mg/dl) | 89.83 ± 7.69 | 88.00 ± 7.21 | 64.57 ± 11.89 | 0.865 | 0.202 | 0.233 |
| Triglycerides (mg/dl) | 57.17 ± 9.08 | 78.33 ± 8.39 | 89.21 ± 6.68 | 0.118 | **0.014** | 0.362 |

[a]All measures are expressed as means ± SEMs. BMI, body mass index; HDL, high-density lipoprotein; LDL, low-density lipoprotein.
[b]P values resulting from pairwise comparison between groups using t test. The Kolmogorov-Smirnov test was used to test the normality of distribution. Skewed data were log transformed for analysis. Statistical difference between pairs of groups was considered when P values were ≤0.05 (bold and underlined).
[c]Statistical comparison of boys and girls distribution between groups by chi-squared test with Yates' continuity correction.
[d]HOMA-IR, homeostatic model assessment for insulin resistance, calculated as fasting [insulin levels (internatonal units per liter) × fasting glucose (millimoles per liter)]/22.5.

adulthood, increasing the rates of early obesity-associated morbidities and mortality (1, 5–7). Notwithstanding, childhood also represents a window of opportunity to reverse these trends through more effective interventions that promote healthier lifestyles and, thus, help prevent excessive weight gain and metabolic inflammation (3, 6).

Classically, obesity in children is the result of an imbalance between energy intake and expenditure, mainly caused by a poor diet and sedentary lifestyle (8). Nonetheless, the gut microbiota has been identified as an additional biological determinant of obesity in recent years, based on findings of comprehensive studies with humans and animal models (9–12). Although the set of microorganisms involved in weight gain and adiposity has yet to be identified in detail, studies show that the obese phenotype can be transmitted to lean individuals by transferring to them the gut microbiota of obese individuals and, also, that the obesogenic features of the microbiota are modulated by diet (12).

Obesity is a risk factor for the onset of insulin resistance (IR), a dysfunctional condition of the glucose metabolism which often leads to pancreatic β cell failure, finally triggering the onset of type 2 diabetes (T2D) (13). Gut microbiota profiling in subjects with IR has outlined associations between certain microbial species and this prediabetic condition (14). Evidence from translational studies also supports a role for gut microbiota alterations in IR development, consistent with parallel alterations in plasma metabolites known to be modulated by the gut microbiota (15). However, conclusions about which specific bacterial species inhabiting the human gut are responsible for disrupting glucose homeostasis are contradictory across human and animal studies (14, 16–18). To address these issues, we aimed to acquire a better understanding of the childhood microbiome and its association with metabolic complications underlying obesity. Consequently, we have analyzed the gut microbiota structure and the cytokine profile in obese children with and without IR and the immune regulatory properties of bacterial species linked to metabolically healthy phenotypes. The results obtained shed light on the role of gut microbes in immune-mediated mechanisms leading to complications underpinning obesity in childhood.

## RESULTS

**Stratification of children as obese and with and without insulin resistance.** The anthropometric features and plasma biomarkers of clinical relevance for obesity and insulin resistance of the subjects included in this study are shown in Table 1. The obese children showed a higher body mass index (BMI) z-score according to the study group classification. They also showed increases in fasting insulin levels and homeostasis model assessment IR (HOMA-IR) indices compared to those of normal-weight children.

**TABLE 2** Immune markers in stool samples of the study children

| Marker analyzed[a] | Value for child group | | | P value of statistical analysis[b] | | |
|---|---|---|---|---|---|---|
| | Lean (n = 16) | Obese (n = 20) | Obese+IR (n = 15) | Lean vs obese | Lean vs obese+IR | Obese vs obese+IR |
| IL-6 | 17.8 (12.4–51.8) | 22.2 (4.0–26.1) | 26.4 (11.8–37.3) | 0.226 | 0.925 | 0.149 |
| IFN-γ | 28.5 (19.8–49.2) | 192.5 (91.0–273.4) | 103.8 (52.0–168.7) | **<0.001** | **0.007** | 0.055 |
| TNF-α | 9.7 (5.7–21.3) | 7.8 (7.8–10.5) | 33.7 (10.8–70.6) | 0.902 | **0.019** | **0.004** |
| MCP-1 | 7.8 (7.8–7.8) | 35.9 (12.1–57.9) | 21.75 (5.9–49.7) | **0.003** | 0.103 | 0.427 |

[a]Values are expressed in nanograms of cytokine per gram of stool and as the medians of respective distributions accompanied by Q1 and Q3 boundaries within parentheses.

[b]P values resulting from pairwise comparison between groups using Wilcoxon rank sum test. Statistical difference between pairs of groups was considered when P values were ≤0.05 (bold and underlined). Multiple-testing correction was applied when more than one pairwise comparison was done at once (Benjamini-Hochberg).

The cutoff for defining children as IR was set at a HOMA-IR index of ≥3.16 (19). Accordingly, children classified as obese showed a lower HOMA-IR index than those classified as obese with IR (obese+IR) ($P < 0.001$). We found no differences in the concentration of fasting glucose in the whole cohort. Regarding the lipid profile, high-density lipoprotein (HDL) cholesterol was notably decreased in both subgroups of obese children, and total cholesterol, as well as triglyceride concentration, was greatly increased in the obese+IR children.

**Inflammatory markers are increased in feces of obese and insulin-resistant children.** The quantification of different proinflammatory cytokines and chemokines in children's stools is shown in Table 2. Obese children showed larger concentrations of gamma interferon (IFN-γ) and monocyte chemoattractant protein 1 (MCP-1) in stools, regardless of IR status (Table 2). Strikingly, higher levels of tumor necrosis factor alpha (TNF-α) were linked only to the IR condition of obese children (Table 2). No meaningful changes in interleukin 6 (IL-6) concentrations were observed among the study groups.

**Microbiota signatures are specific for obesity and obesity with insulin resistance.** The alpha diversity of the fecal microbiota did not reveal notable changes in the number of species or their abundances, either between normal-weight and obese children or between obese children with and without IR. The beta diversity analysis using the Bray-Curtis dissimilarity index indicated that the structure of the fecal microbiota of normal-weight children differed from that of obese ones (permutational multivariate analysis of variance [PERMANOVA] = 2.08; $P = 0.003$) (Fig. 1). No meaningful associations of microbiota profiles with age (PERMANOVA = 1.05; $P = 0.361$) or sex (PERMANOVA = 0.82; $P = 0.714$) of children were intuited. A principal-coordinate analysis (PCoA) indicated that the multidimensional information compiled in principal coordinate (PC) 1 seemed to discriminate the gut microbiota of normal-weight children from that of obese children regardless of their IR status ($P = 0.051$); however, the multivariate information compiled in the PC2 enabled us to discriminate the microbiota of obese+IR children from that of the other two groups ($P < 0.001$), thus suggesting the presence of microbial features specifically linked to this condition. To disclose the distinctive features, we performed a nonparametric permutation-based test in a pairwise fashion. We retrieved a total of 7 operational taxonomic units (OTUs) that had a significantly higher abundance in normal-weight children, whereas only 1 was more abundant in the obese+IR group (Fig. 2). Most of the OTUs associated with the lean phenotype in children were bacterial species belonging to the family *Lachnospiraceae* of the phylum *Firmicutes* (6 out of 7). This was also the case for OTU266, but this was found to be increased in obese+IR children. Five OTUs were identified at the genus level using SINA aligner and the SILVA 16S reference database, and three of them were identified as *Blautia* species (OTU13, OTU299, and OTU662). Further identifications at the species level for OTUs related to *Blautia* were retrieved from BLAST-based comparisons against the NCBI 16S reference database and DADA2 algorithms using the RefSeq+RDP database (see Materials and Methods). We determined that *Blautia luti* (OTU13) and *Blautia wexlerae* (OTU299 and OTU662), as well as *Eubacterium hallii*

# Principal Coordinate Analysis – PCoA
## (Bray-Curtis dissimilarity)

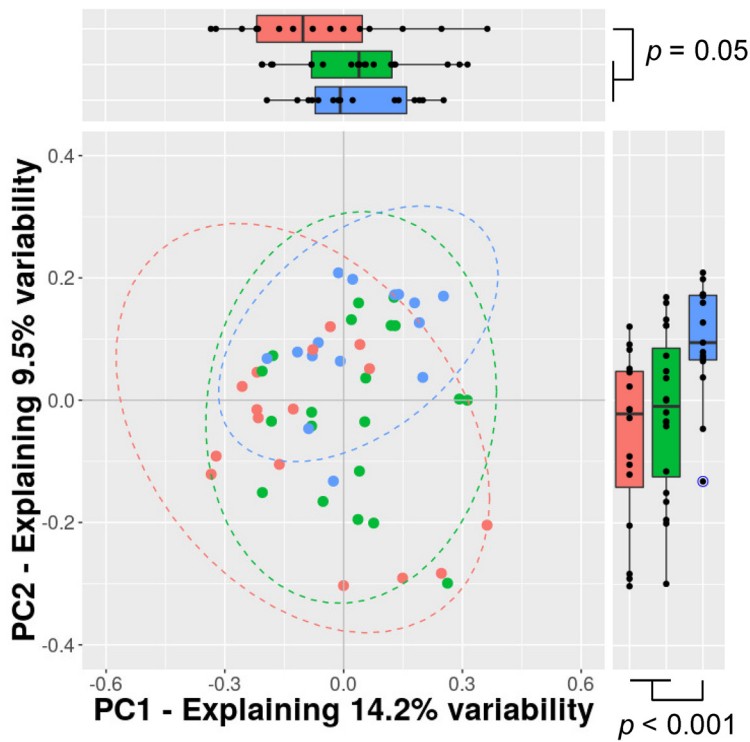

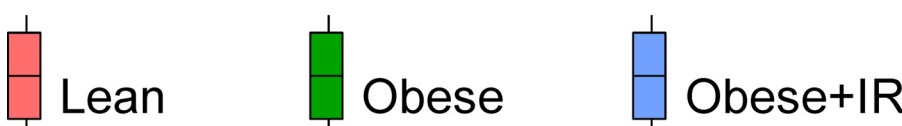

**FIG 1** Beta diversity of the gut microbiota profiles from normal weight and obese children. A principal-coordinate analysis (PCoA) of dissimilarities among samples is shown, with marginal boxplots disclosing the distribution of the two most informative principal coordinates (PC) of the multidimensional analysis. Blue data points correspond to outliers. The color key for the scatter- and boxplots indicate the study groups of children. PERMANOVA values = 2.08 and 1.68 ($P < 0.005$) for comparison using two-group (lean versus obese) and three-group (lean versus obese versus obese+IR) configurations, respectively.

(OTU680), were the species likely associated with normal-weight children. The performance of our novel permutation-based approach was preliminarily evaluated by comparing the differentially abundant OTUs predicted by LEfSe and DESeq2. We found consensus signals among the three methods for 4 out of the 7 OTUs detected by our approach (Table 3). In this way, we confirmed that the *Blautia*-associated OTUs (OTU13, OTU299, and OTU662), as well as the *Eubacterium hallii*-associated OTU (OTU680), likely constitute microbial species showing a reduced abundance in obese children.

**Gut microbiota signatures correlate to inflammatory markers relevant for obesity and IR.** To identify the set of potential intestinal bacteria contributing to obesity or to obesity with metabolic complications, namely, IR, we established correlations between abundances of selected OTUs and fecal immune markers that discriminate among the study groups, including MCP-1, IFN-$\gamma$, or TNF-$\alpha$ (Fig. 3A and B). We found that a total of three OTUs (OTU13, OTU98, and OTU662) exhibited significant negative correlations with TNF-$\alpha$ values and that of these, OTU13 (*B. luti*; $\rho = -0.37$ and

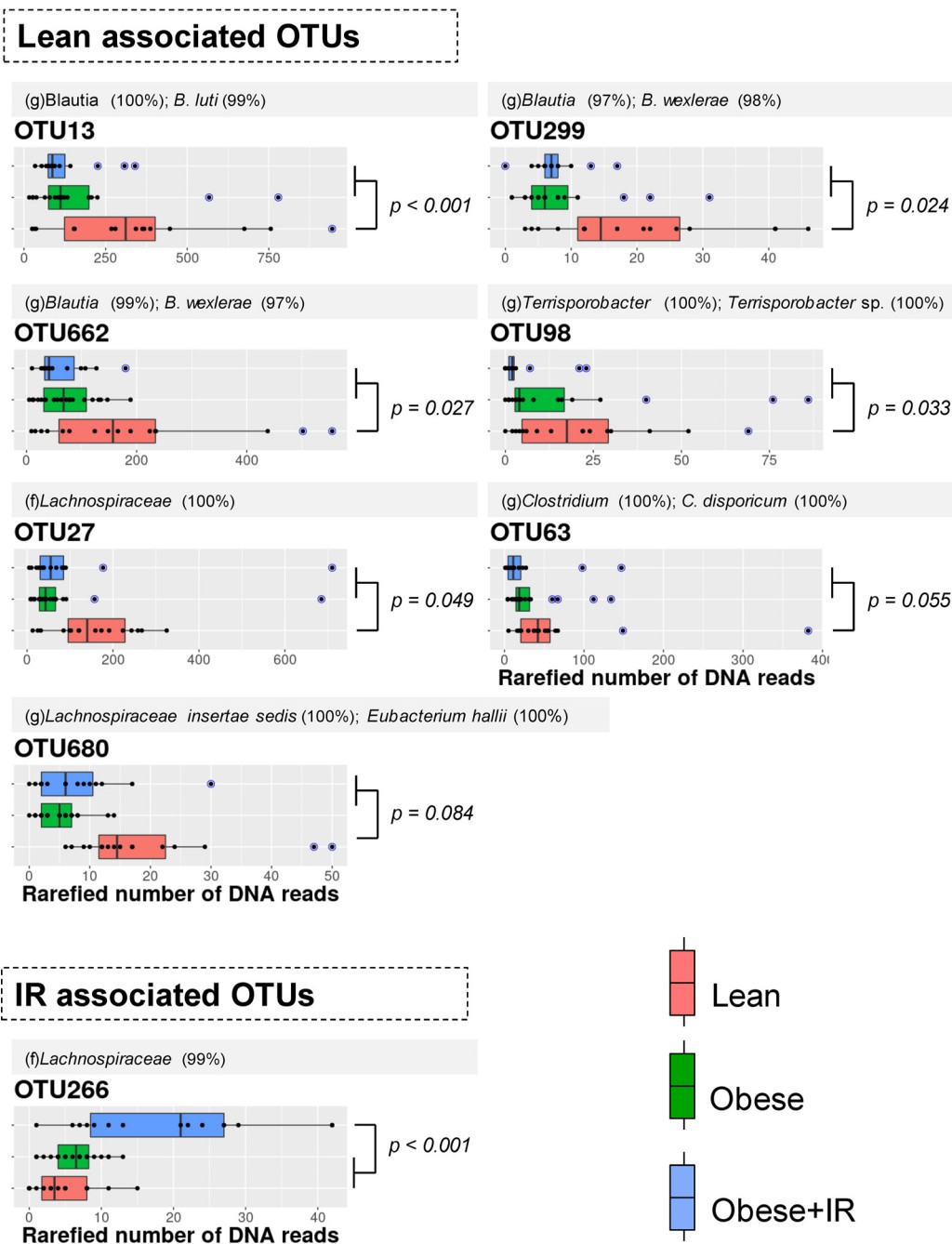

**Lean associated OTUs**

(g)Blautia (100%); *B. luti* (99%)
**OTU13**
*p < 0.001*

(g)*Blautia* (97%); *B. wexlerae* (98%)
**OTU299**
*p = 0.024*

(g)*Blautia* (99%); *B. wexlerae* (97%)
**OTU662**
*p = 0.027*

(g)*Terrisporobacter* (100%); *Terrisporobacter* sp. (100%)
**OTU98**
*p = 0.033*

(f)*Lachnospiraceae* (100%)
**OTU27**
*p = 0.049*

(g)*Clostridium* (100%); *C. disporicum* (100%)
**OTU63**
*p = 0.055*
**Rarefied number of DNA reads**

(g)*Lachnospiraceae insertae sedis* (100%); *Eubacterium hallii* (100%)
**OTU680**
*p = 0.084*
**Rarefied number of DNA reads**

**IR associated OTUs**

(f)*Lachnospiraceae* (99%)
**OTU266**
*p < 0.001*
**Rarefied number of DNA reads**

Lean

Obese

Obese+IR

**FIG 2** OTUs with differential abundances among child groups. A permutation-based test was performed between pairs of groups to disclose taxonomic features associated with different conditions. Differential abundance was assumed when corrected *P* values were <0.1 (FDR test). Respective *P* values supporting the biomarker discovery approach are shown accordingly along with a color key for boxplots. Blue data points indicate outliers. Taxonomic identification of OTUs was based on SINA aligner using default parameters; shown are reliable identifications based on the SILVA database with the level of sequence identity (within paretheses) against their last common ancestor (lca). If genus was identified, we further identified the OTU at the species level in a BLAST-based approach (top alignment score and length) against the NCBI 16S reference database and classifier algorithms implemented in DADA2 with the combined RefSeq+RDP database. The sequence identity is shown. f, family; g, genus.

*P* = 0.009) and OTU662 (*B. wexlerae*; $\rho = -0.36$ and *P* = 0.012) showed the strongest correlations, suggesting that the higher their abundance, the lower the TNF-$\alpha$ levels. The remaining OTU, OTU299, also identified as *B. wexlerae*, exhibited a similar trend ($\rho = -0.26$ and *P* = 0.074). Moreover, we found that five OTUs negatively correlated with IFN-$\gamma$ values. These were OTU13 ($\rho = -32$ and *P* = 0.023), OTU27 ($\rho = -41$ and

**TABLE 3** Permubiome performance against common microbiome biomarker discovery tools[a]

| Reference algorithm (no. of discriminated OTUs) | Permubiome OTUs (n = 8) | | |
| --- | --- | --- | --- |
| | Detected[b] | Not detected[c] | P value[d] |
| LEfSe (25) | 6 [5 + 1] (LDA = 3.38 ± 0.38) | 19 (LDA = 3.09 ± 0.43) | 0.147 |
| DESeq2 (38) | 4 [4 + 0] (log$_2$ FC = 1.45 ± 0.32) | 34 (log$_2$ FC = 1.62 ± 1.71) | 0.627 |

[a]Consensus OTUs detected were OTU13, OTU299, OTU662, and OTU680. LEfSe, linear discriminant analysis effect size; LDA, linear discriminant analysis; FC, fold change.
[b]Number of OTUs equally detected to be associated with lean and obese phenotypes by permubiome and the reference algorithms. Square brackets indicate those associated with lean and obese conditions.
[c]Number of OTUs detected to be associated with lean and obese phenotypes by reference algorithms but not by permubiome.
[d]P values resulting from pairwise comparison between scores of OTUs detected and not detected by permubiome using Student t test (normally distributed data). The criterion used for selecting OTUs (P ≤ 0.05) in LEfSe analysis was an LDA of ≥2, whereas absolute log$_2$ FC was the criterion used in DESeq2.

$P = 0.004$), OTU63 ($\rho = -0.38$ and $P = 0.008$), OTU662 ($\rho = -0.33$ and $P = 0.022$), and OTU680 (*E. hallii*; $\rho = -0.43$ and $P = 0.002$). Overall, these findings are in agreement with associations established between child groups and the gut microbiota signatures reported previously, except for OTU662. Finally, we detected one negative correlation between MCP-1 values and OTU63 abundance ($\rho = -0.35$ and $P = 0.013$).

***Blautia* spp. linked to normal-weight children show anti-inflammatory properties.** An *in vitro* experiment was designed to provide further evidence of the role of *Blautia* spp., which were abundant in normal-weight children but depleted in obese children, as possible regulators of obesity-associated inflammation. We used the ratio of IL-4 to IFN-γ or TNF-α as an index of the anti-inflammatory response of peripheral blood mononuclear cells (PBMCs) when exposed to different *Blautia* species, such as *Blautia luti* DSM 14534 and *Blautia wexlerae* F15, associated with a healthy metabolic phenotype in our observational study in children. The effects were compared to those of a strain of the species *Bacteroides vulgatus* (see Materials and Methods), corresponding to the identity of the OTU1 (SINA = *Bacteroides* 100%; BLAST = *Bacteroides vulgatus* 100%). This OTU was the most abundant in the children studied and show no differences among groups ($P < 0.191$ and false-discovery rate [FDR] = 1.0) and no correlations with MCP-1, IFN-γ, and TNF-α concentrations ($\rho = 0.12$ and $P = 0.419$, $\rho = 0.24$ and $P = 0.086$, and $\rho = 0.14$ and $P = 0.342$, respectively). *Blautia* species increased the anti-inflammatory cytokine ratio compared to *B. vulgatus* BAC-CCC-2 (Fig. 3C). Moreover, most of differences established among groups were strongly influenced by IL-4/IFN-γ ratio (Fig. 3C).

## DISCUSSION

This study has identified gut microbiota signatures underlying obesity and insulin resistance, a critical metabolic dysfunction often present in obese children which also represents a risk for the development of T2D in adulthood. Our study has also established new functional links between the microbiota species depleted in obese children and the increased intestinal inflammatory markers that could partly account for insulin resistance.

Low-grade chronic inflammation is associated with obesity and causally linked to the development of IR in these subjects, which may subsequently progress to chronic metabolic disease, like T2D (20). Although adipose tissue is recognized as a major contributor to inflammation and metabolic dysfunction during obesity, we now know that this process affects different organs, including the muscle, liver, brain, and gut (21). The intestinal microbes adapted to unhealthy diets are viewed as an additional source of inflammatory signals that contribute to metabolic inflammation in obesity (22). Our findings indicated that obese children, with or without IR, exhibited an elevated intestinal inflammatory tone, supported by the accumulation of proinflammatory mediators in the intestine, such as the cytokines IFN-γ and TNF-α and the chemokine MCP-1. This is consistent with previous findings obtained in animal models of obesity (21). Accordingly, when microbiota exposed to either high saturated fat or fish oil are transferred to new recipients, it is found that the interactions between the unhealthy diet and the gut microbiota exacerbate metabolic inflammation. Also, studies show that

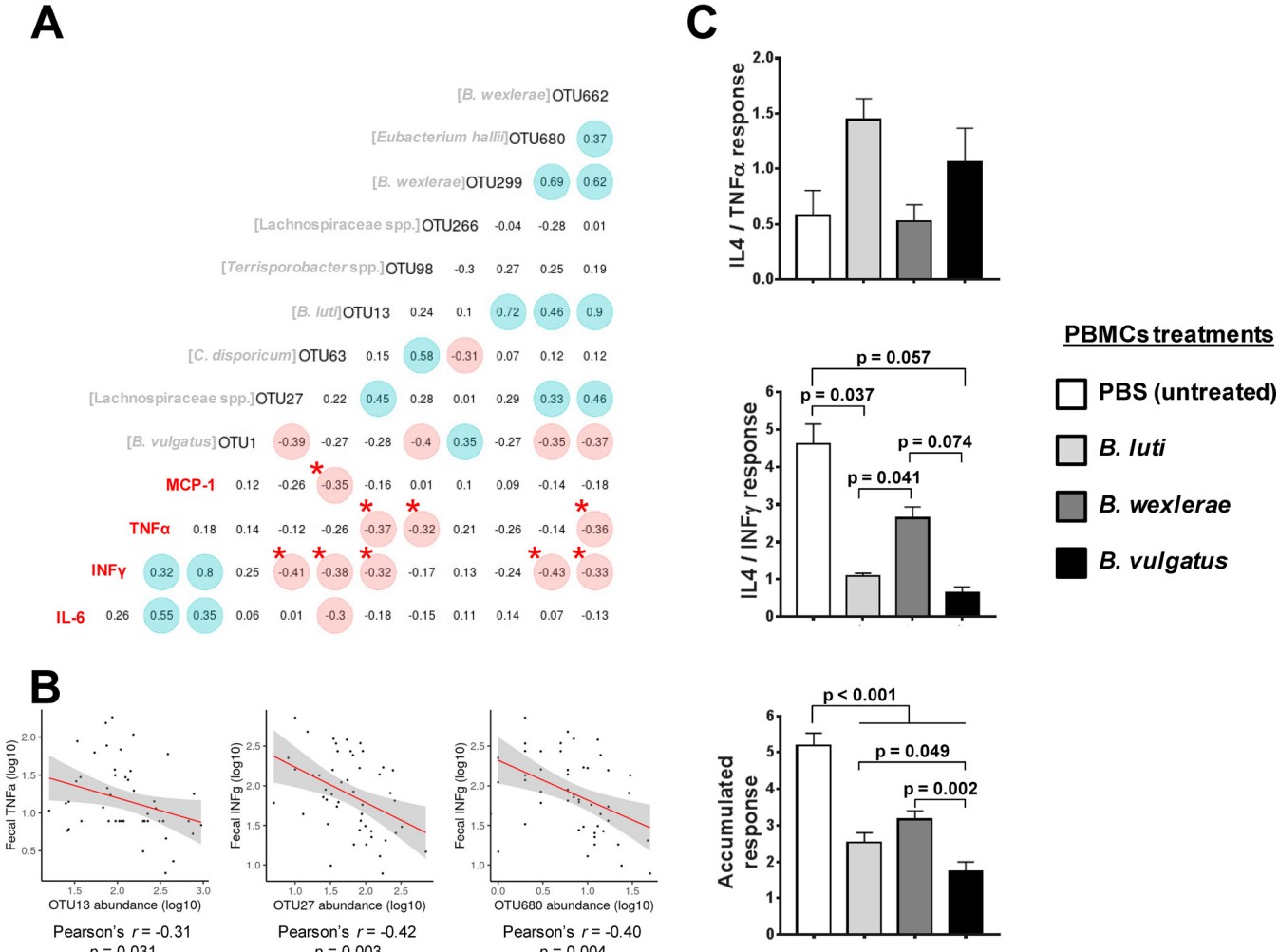

**FIG 3** Links between gut microbiota and inflammatory markers. (A) Spearman correlations between the OTUs selected and the proinflammatory markers analyzed. The strongest correlations supported by larger $\rho$ values are highlighted accordingly (positive are shaded blue and negative are shaded red). Asterisks indicate correlation with corrected $P$ values of $\leq 0.05$. (B) Scatterplots of selected correlations between fecal cytokines and OTUs exhibiting top $\rho$ values. The axes were plotted as log-transformed data, and linear correlations (slope as red line) as well as confidence intervals (gray shadow) are drawn in all cases. (C) Anti-inflammatory to proinflammatory cytokine ratios (IL-4/IFN-$\gamma$ and IL-4/TNF-$\alpha$ individually represented) detected upon exposure of PBMC cultures to different bacterial stimuli *in vitro*. The botton barplot represent the sum of both cytokine ratios. Differences between the effects of different bacterial stimuli established by one-way ANOVA with Bonferroni correction for multiple testing at a $P$ value of <0.05 are shown in bar graphs.

specifically microbially induced MCP-1 enhances macrophage accumulation in white adipose tissue (WAT), promoting inflammation and insulin resistance, regardless of obesity (23). This and previous studies also suggest that the inflammatory process, resulting from the synergic role between a high-fat diet and diet-induced dysbiosis, disrupts gut barrier integrity, facilitating the translocation of inflammatory bacterial products (e.g., lipopolysaccharide [LPS]) to the systemic circulation and, for example, boosting the serum's ability to activate innate immunity receptors, like Toll-like receptor 4 (TLR4) and cytokine gene expression, like *Tnfa*, in both adipocytes and macrophages (20, 23). It is well known that the cytokine TNF-$\alpha$ is directly involved in IR via the impairment of insulin-mediated cell signaling in metabolic organs, such as adipose tissue (24).

Furthermore, the inflammatory milieu of the obese intestine may also reflect imbalances in immune cell populations with a proinflammatory profile, as shown in animal studies (25), which participate in the immune cell traffic from the gut to other lymphoid organs such as the white adipose tissue, triggering inflammation (21).

Despite manifold attempts made to identify the gut microbiota signatures predic-

tive of obesity and T2D, the precise set of microorganisms contributing to or triggering such conditions has not been identified as yet. It is also likely that there is not just one single microbiota pattern linked to such complex conditions, and thus, we should classify subjects according to more precise features of their metabolic phenotype. With this purpose, we have used massive sequencing of V4-V5 amplicons of the bacterial 16S rRNA gene to analyze the intestinal microbiota of lean and obese children with and without IR. Overall, we found differences in gut microbial composition between lean and obese children by using beta diversity metrics. Using a novel permutation-based approach for microbiome biomarker discovery, we found differential abundances in several OTUs associated predominantly with the metabolically healthy phenotype, whereas only one OTU, lacking definite taxonomic identification, was associated with the obese+IR phenotype. A common signature of the lean-child phenotype is that most of the associated OTUs (6 out of the 7 OTUs) were classified as members of the *Lachnospiraceae* family of *Firmicutes*. Moreover, three of them were taxonomically identified as members of the genus *Blautia* and the species *B. luti* (OTU13) and *B. wexlerae* (OTU299 and OTU662), which showed the strongest associations. It is noteworthy that the difference in the reduced abundance of these OTUs increased from lean to obese subjects and to obese+IR subjects, suggesting an association between their depletion and metabolic phenotype deterioration. Indeed, *B. wexlerae* has previously been associated with a nonobese phenotype in adults (26), and a Japanese population-level gut microbiota assessment negatively correlated *Blautia* genus abundance with the visceral fat area, an adiposity biomarker for risk of cardiovascular and metabolic disease (27). Remarkably, a clinical trial with T2D patients revealed that combined treatment with metformin and a Chinese herbal formula improved the glucose and lipid profile in T2D subjects concomitantly with an increase in *Blautia* species (28). These results further support our predictions and provide evidence for the potential role of *Blautia* species in the maintenance of a metabolically healthy phenotype and the management of obesity, IR, and T2D. To advance in our understanding of the role these bacterial species play in obesity and IR, we characterized the potential anti-inflammatory effects of two strains of *B. luti* and *B. wexlerae*. We observed that *B. wexlerae* F15 reduced the ratio of IFN-$\gamma$ to IL-4 to a larger extent, while *B. luti* DSM 14534 seemed to reduce the ratio of TNF-$\alpha$ to IL-4 upon exposure to PBMCs instead, compared to the effects of *B. vulgatus* BAC-CCC-2, another commensal species equally present in all child groups. These findings suggest a mechanism whereby *Blautia* species could contribute to maintaining glucose homeostasis (28). This could be mediated by their ability to balance the proinflammatory and anti-inflammatory mediators of the immune response, whose dysregulation is well known to impair the insulin signaling inside the cells (13, 24).

*Blautia* species (*Clostridium* cluster XIVa) are also well recognized as part of the butyrate-producing bacteria of the intestinal microbiota. This is a bacterial metabolite that could account for the beneficial roles of these bacteria in glucose metabolism and obesity-associated inflammation (29, 30). Intriguingly, a recent report from the integrative Human Microbiome Project Consortium (iHMP), which followed prediabetic individuals for 4 years, indicated that an abundance of *Blautia* genus negatively correlated with hippuric acid levels and positively correlated with plasma glucose concentration measured by insulin suppression test to determine IR (31). Similarly, Egshatyan and coworkers also reported a correlation of *Blautia* genus abundance with altered glucose tolerance (32). Although our results contradict those previously published, the former studies only reported associations between *Blautia* species and glucose metabolic dysfunction and did not support causation. Furthermore, differences depending on the specific *Blautia* species involved could not be disregarded. For example, a proinflammatory response was attributed to other *Blautia* species, such as *Blautia coccoides*, based on the secretion of TNF-$\alpha$ and IL-10 by PBMCs *in vitro* (33). In contrast, we demonstrated that *B. luti* and *B. wexlerae* exerted anti-inflammatory effects on PBMCs. These findings suggest that differences at the species level could lead to different associations with healthy or impaired glucose metabolism in humans. Further studies

are, therefore, warranted to investigate this hypothesis in depth using *in vivo* study models in order to draw definitive conclusions about the potential and differential role of *Blautia* species in the management of obesity and its metabolic complications. For example, one of our previous intervention trials in overweight subjects indicates that intake of a wheat bran extract enriched in arabinoxylans boosts the proportion of *Blautia* species (34, 35), suggesting that it is possible to design diet-based interventions to enrich the gut ecosystem with these bacterial species depleted in obese children. In spite of this, preclinical and clinical intervention trials with *B. luti* and *B. wexlerae* strains would be needed to definitively demonstrate their potential protective effect against obesity and prediabetic states.

**Conclusions.** This study indicated that *B. luti* and *B. wexlerae* species could contribute to the maintenance of intestinal immune homeostasis in metabolically healthy subjects and that their depletion is associated not only with obesity but also with metabolic complications such as insulin resistance and related inflammatory markers. This could help to inform future gut microbiota-based interventions applicable to childhood, taking advantage of this window of opportunity for metabolic disease prevention.

## MATERIALS AND METHODS

**Study subjects, sampling, and clinical assessments.** Children were recruited between 2013 and 2014 at the outpatient clinic of the Dr. Peset University Hospital (Valencia, Spain). The study included 51 children (26 boys and 25 girls) between the ages of 5 and 17 years and of Caucasian race. Anthropometry measurements (weight and height) were obtained using standardized clinical protocols. Thirty-five subjects were obese, with a body mass index z-score of ≥2, and were attending the outpatient clinic of the pediatrics service, while 16 were children with normal nutritional status from the local population who were visiting primary care pediatricians for routine health check-ups and were unrelated to the patients. Exclusion criteria included any known genetic disorder, syndrome, or disease that could influence dietary intake, body composition and fat distribution, endocrine disorders, medication use, or unusual dietary habits (e.g., vegetarianism). None of the children had chronic diseases or suffered from inflammatory conditions, and they had not previously participated in a structured exercise or weight loss program. The children were clinically evaluated to detect any acute disease or infectious illness. Written informed consent was obtained from all parents, and verbal agreement witnessed and formally recorded was obtained from all children. The study was approved by the Ethical Committee of the Dr. Peset University Hospital.

**Biochemical parameter assessment in blood from children.** Blood samples were collected after 12-h fasting. Biochemical characterization tests using automated direct methods (Architect C16000; Abbott Clinical Chemistry, Wiesbaden, Germany) were used to analyze serum glucose, insulin, total cholesterol, low-density lipoprotein cholesterol, high-density lipoprotein cholesterol, and triglycerides in the central laboratory of the Dr. Peset University Hospital. The homeostasis model assessment index (HOMA) was used as surrogate marker to determine IR by employing the following formula: fasting insulin levels (international units per liter) × fasting glucose (millimoles/liter)/22.5. We defined insulin resistance as an HOMA-IR index of ≥3.16 (19). According to this value, 16 obese children were considered to have IR. A fecal sample was obtained the same day as analytical procedures for intestinal microbiota assessment. Respective aliquots for fecal cytokine and gut microbiota assessments were immediately prepared before freezing at −80°C.

**Inflammatory marker quantification in stools of children.** The concentrations of soluble proinflammatory markers (the cytokines interleukin 6 [IL-6], gamma interferon [IFN-γ], and tumor necrosis factor alpha [TNF-α] and monocyte chemoattractant protein 1 [MCP-1]) on stool supernatants were measured by enzyme-linked immunosorbent assay (ELISA) following the manufacturer's instructions (BioLegend, San Diego, CA). Briefly, 1 g of stool was diluted 1:10 (wt/vol) in sterile 1× phosphate-buffered saline (PBS; pH 7.4) and homogenized by gentle vortexing, and supernatants were collected by centrifugation at 4°C and 16,000 × *g* for 10 min. Stool supernatants were diluted 1:10 in 1× assay diluent (BioLegend, San Diego, CA) and used for ELISAs (stool final dilution, 1:100 [wt/vol]). Data are expressed in nanograms of cytokine per gram of stool. Values below the limit of detection for each ELISA were adjusted to that limit.

**Microbiota analysis.** Approximately 150-mg stool aliquots were processed for DNA isolation using the QIAamp DNA stool minikit (Qiagen, Hilden, Germany) following the manufacturer's instructions. A diluted aliquot of the fecal DNA was prepared at 20 ng/μl in sterile and nuclease-free water for PCRs. Approximately 20 ng of DNA (1 μl of diluted DNA) was used to amplify the V4-V5 hypervariable regions from the bacterial 16S rRNA gene by a 25-cycle PCR program consisting of the following steps: 95°C for 15 s, 40°C for 30 s, and 72°C for 20 s. The PCR was performed using Phusion high-fidelity *Taq* polymerase enzyme (Thermo Scientific) and 6-mer barcoded primers, which target a wide range of bacterial 16S rRNA genes: S-d-Bact-0563-a-S-15 (AYTGGGYDTAAAGNG) and S-d-Bact-0907-a-A-20 (CCGTCAATTYMTTTRAG TTT) (36). Dual-barcoded amplicons consisted of approximately 400-bp fragments purified from triplicate reactions per sample using the Illustra GFX PCR DNA and gel band purification kit (GE Healthcare, Little Chalfont, UK). Amplicon DNA was quantified using the Quant-iT PicoGreen double-stranded DNA

(dsDNA) assay kit (Thermo Fisher Scientific, Waltham, MA). All samples were multiplexed by combining equimolar quantities of amplicon DNA (150 ng per sample) and sequenced in an Illumina MiSeq platform with $2 \times 300$ PE configuration (Eurofins Genomics, Ebersberg, Germany).

The raw amplicon DNA sequencing data were delivered in FASTQ files, and pair ends with quality filtering were assembled using *Flash* software (37). Sample demultiplexing was carried out using the sequence information from respective DNA barcodes and the Mothur v1.32.1 suite of analysis (38). After barcode/primer removal, the sequences were processed for chimera detection using the Uchime algorithm (39) and the reference set of 16S sequences from the SILVA database (40) (release 110) implemented in the respective version of the Mothur package used. The taxonomy composition analysis was assessed at the operational taxonomic unit (OTU) level with a rarefied subset of 12,000 sequences per sample, set as the minimum coverage obtained, and randomly selected after multiple shuffling (10,000×). The clustering of sequences was at 97% sequence identity using the uclust algorithm implemented in USEARCH v8.0.1623 (41). Different alpha diversity descriptors such as Chao's richness, Simpson's evenness, Simpson's reciprocal index, and observed richness were evaluated on the OTU abundance data using qiime v1.9.1 (42). Statistical appraisal of the above-mentioned parameters between groups was estimated with Wilcoxon rank sum test in R v3.5. The microbial community structure and changes between groups, regardless of phylogenetic relationships among OTUs, were evaluated through beta diversity approaches using Bray-Curtis dissimilarity index followed by permutation-based statistical assessment with PERMANOVA, all implemented in qiime v1.9.1. Lean/obese phenotype, age, and sex were assessed as variables explaining microbial community structure. Graphical exploration of similarities between groups was performed through the principal-coordinate analysis (PCoA) multidimensional scaling approach also implemented in qiime v1.9.1.

**Data analysis of 16S sequences for microbiota-based biomarker discovery.** The differential abundance of OTUs between pairs of groups was determined as the difference of the normalized DNA read count medians observed between them, here referred to as $\Delta_{ij}$. Briefly, the significance of $\Delta_{ij}$ for each OTU feature was then evaluated using a permutation-based test (1,000×), thus allocating randomly all observations per OTU into groups, and then the $\Delta_{ij}$ was recalculated after every iteration. The distribution of $\Delta_{ij}$ across all permutations was observed to follow a normal distribution, and then z-scores were calculated for the observed $\Delta_{ij}$ and for those resulting from multiple permutations independently for every OTU category. The cumulative probability for the observed $\Delta_{ij}$ was then calculated taking into account both sides of the distribution to distinguish over- and underrepresented OTUs in the case group (lean). Finally, multiple-testing correction of the resulting $P$ values was done using the Benjamini-Hochberg method (or false-discovery rate [FDR]) over the hundreds of comparisons performed at once for OTU categories selected to be more abundant ($n = 320$). The OTUs were considered to have differential abundance between groups when corrected $P$ values were ≤0.1. The above-described algorithms and accompanying functions were built in R v3.6 and compiled in an R package called permubiome (see "Availability of data and material" below). The SINA aligner (43) and BLAST-based searching (top scoring) (https://blast.ncbi.nlm.nih.gov/Blast.cgi) using, respectively, the SILVA and NCBI 16S reference databases were used to assign the most probable taxonomy identity of the selected OTU sequences. Briefly, the OTUs that could be identified by SINA aligner at the genus level were further identified at the species level with the BLAST-based approach. Top hit selection was based on the taxonomy score (TS): TS $= \log_{10}$(alignment score $\times$ sequence identity $\times$ alignment length). Additionally, we used the classifier algorithm implemented in DADA2 v3.10 (44) with the combined RefSeq+RDP reference database to further assess the OTU identities. A preliminary analysis was also carried out to compare the performance of permubiome with those of two other conventional methods used for biomarker discovery, LEfSe (45) and DESeq2 (46), to validate our methodology.

**Immunomodulatory effects of bacterial isolates *in vitro*.** The type strain *Blautia luti* DSM 14534 was obtained from the Leibniz Institute DSMZ-German Collection of Microorganisms and Cell Cultures (https://www.dsmz.de). The other two bacterial strains (*Blautia wexlerae* F15 and *Bacteroides vulgatus* BAC-CCC-2) were isolated from feces of the study donors. Briefly, fresh stool samples were collected, kept under anaerobic conditions (AnaeroGen; Oxoid), and stored at 4°C until processing (up to a maximum of 12 h after collection). Feces were diluted 1:10 (wt/vol) in oxygen-depleted (reduced) 1× PBS containing 0.05% L-cysteine (rPBS), homogenized and serially diluted in rPBS, and plated in Gifu anaerobic agar (GAM broth; Himedia) and in Schaedler agar (Scharlau, Spain). After 72 h of incubation at 37°C in an anaerobic chamber (Bactron-300; Shel-lab), single colonies were isolated and identified by PCR using the universal primers 27F (5′-AGA GTT TGA TCC TGG CTC AG-3′) and 1401-R (5′-CGG TGT GTA CAA GAC CC-3′), which amplify the nearly full 16S rRNA gene. PCR products were cleaned with Illustra GFX PCR DNA and gel band purification kit (GE Healthcare, Chicago, IL) and sequenced by Sanger technology in an ABI 3730XL sequencer (STAB-VIDA, Caparica, Portugal). The taxonomy identity was established by BLASTn comparisons (https://blast.ncbi.nlm.nih.gov/Blast.cgi) against the nonredundant 16S NCBI database. For immunological assays, the bacterial strains were growth in the most appropriate medium (GAM broth for *Blautia* spp. and Shaedler broth for *B. vulgatus*) as described above. Afterwards, bacterial cells were collected by centrifugation and washed with sterile rPBS buffer and aliquots containing 20% glycerol were immediately frozen in liquid nitrogen until used.

PMBCs were isolated from whole blood of three healthy donors by density gradient centrifugation using Ficoll (Ficoll Paque-Plus; Bioscience) according to the manufacturer's instructions. PBMCs were then washed in lysis buffer for red blood cells (Miltenyi Biotec, Bergisch Gladbach, Germany) and diluted in Gibco RPMI 1640 (Thermo Fisher Scientific, Waltham, MA) containing 10% fetal calf serum, 100 $\mu$g/ml of streptomycin (Sigma), 100 U/ml of penicillin (Sigma), and 4.5 g/liter of L-glutamine (Sigma). Cells were maintained in a humidified atmosphere at 37°C with 5% $CO_2$ at $10^5$/ml in 24-well polystyrene plates

(Corning, Edison, NY). Isolated PBMCs were stimulated with live bacterial suspensions at a final concentration of $10^6$ CFU/ml. Treatments with 1 $\mu$g/ml of bacterial LPS (Sigma) and sterile 1$\times$ PBS were used as positive and negative controls, respectively, for production of proinflammatory cytokines. After 24 h of stimulation, cells were collected and centrifuged for 5 min at 500 $\times$ $g$ and 4°C and diluted at $10^6$ cells/ml in fluorescence-activated cell sorting (FACS) buffer (sterile 1$\times$ PBS with 0.2% [wt/vol] bovine serum albumin [BSA]) to characterize the immunomodulatory properties of the bacteria through flow cytometry. Classic monocyte levels (CD14$^+$ and CD16$^-$) were determined by staining cellular surface markers with fluorescent dye-labeled mouse monoclonal antibodies: phycoerythrin (PE)-conjugated anti-CD14 (130-110-577; Miltenyi Biotec, Bergisch Gladbach, Germany) and fluorescein isothiocyanate (FITC)-conjugated anti-CD16 (130-106-761; Miltenyi Biotec). Additionally, intracellular cytokines were stained in permeabilized and fixed cells (fixation/permeabilization solution kit; BD Bioscience) with fluorescent dye-labeled mouse monoclonal antibodies: allophycocyanin (APC)-conjugated anti-IFN-$\gamma$ (catalog no. 130-109-312; Miltenyi Biotec), PE-Vio615-conjugated anti-IL-4 (catalog no. 130-107-199; Miltenyi Biotec), and PE-Vio770-conjugated anti-TNF-$\alpha$ (catalog no. 130-098-891; Miltenyi Biotec). Cells were analyzed with BD LSR Fortessa (Becton, Dickinson Biosciences, Franklin Lakes, NJ). The data were analyzed using BD FACS DIVA software v7.0 (Becton, Dickinson Biosciences). The response of PBMCs under different bacterial stimuli was considered anti-inflammatory according to the ratio of cells positively labeled with the key cytokines. These were IL-4/IFN-$\gamma$ (indicator of Th2/Th1 balance) and IL-4/TNF-$\alpha$ (indicator of Th2 response/proinflammatory innate immune response). This approach has been extensively used in previous studies to determine the balance between anti-inflammatory and proinflammatory responses triggered by different stimuli (47–49). IL-4 was primarily considered as anti-inflammatory cytokine instead of IL-10 since IL-10 production is under the control of IL-4 (50–52).

**Statistical analyses.** For blood biochemical assays, the $P$ values resulting from pairwise comparison between groups were obtained by using $t$ test and Welch's correction with prior evaluation of data with the Kolmogorov-Smirnov test to test the normality of distributions. Fecal cytokine assessment was also subjected to normality testing on respective cytokine data prior to comparisons. Consequently, Wilcoxon rank sum (unpaired) test was used to evaluate differences between groups, with Benjamini-Hochberg correction when multiple comparisons were made. Correlations among normalized DNA read counts of differentially abundant OTUs and cytokine concentrations in all samples analyzed were performed by estimating the Spearman's rho ($\rho$) parameter. Differences in the cytokine responses of PBMCs obtained from the three different donors against the bacterial strains were estimated by one-way ANOVA with Bonferroni correction.

**Availability of data and material.** The raw data set of demultiplexed fastq sequences was submitted to the European Nucleotide Archive (ENA), and they can be freely accessed under project number PRJEB37005. The functions for the biomarker discovery approach were compiled with R v3.6 into the package called permubiome, which is distributed under the GPL-3 license and freely downloaded from the Comprehensive R Archive Network (CRAN) at https://cran.r-project.org/ or from https://github .com/alfbenpa/permubiome. Partial 16S rRNA sequences obtained and assembled from *B. vulgatus* BAC-CCC-2 and *B. wexlerae* F15 isolates are publicly available at the ENA (Bioproject PRJEB32485) under accession numbers LR590078 and LR590079, respectively.

## ACKNOWLEDGMENTS

This study was supported by grant AGL2017-88801-P from the Ministry of Science, Innovation and Universities (MCIU; Spain) and the European Union's Seventh Framework Program under grant agreement no. 613979 (MyNewGut). The contracts of A.B.-P. and E.M.G.D.P. were supported by the MyNewGut project. The FPI grant for I.L.-A. from MCIU (Spain) is fully acknowledged.

We thank the parents and relatives of children recruited in the present study.

Y.S. and P.C.-F. designed and coordinated the present study. A.M.-P. assisted in the collection of samples from children. P.C.-F. was primarily responsible for clinical and biochemical characterization of participants and their samples. A.B.-P. conducted the gut microbiota experimental procedures, massive data analysis, and determination of inflammatory molecules on stools. E.M.G.D.P. isolated and cultivated the bacterial strains. I.L.-A. performed the immunological analyses on PBMCs. All authors participated in paper drafting and approved the final version.

We declare that we have no conflict of interest.

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
