## [Reviewer comments · mSystems]

Depletion of *Blautia* species in the microbiota of obese children relates to intestinal inflammation and metabolic phenotype worsening

Alfonso Benítez-Páez, Eva Gómez del Pulgar, Inmaculada López-Almela, Ángela Moya-Pérez, Pilar Codoñer-Franch, and Yolanda Sanz

Corresponding Author(s): Alfonso Benítez-Páez, Institute of Agrochemistry and Food Technology Institute, Spanish National Research Council (IATA-CSIC)

Review Timeline:

Submission Date:	December 10, 2019
Editorial Decision:	January 14, 2020
Revision Received:	February 27, 2020
Editorial Decision:	February 27, 2020
Revision Received:	February 28, 2020
Editorial Decision:	February 28, 2020
Revision Received:	March 3, 2020
Editorial Decision:	March 3, 2020
Revision Received:	March 5, 2020
Accepted:	March 6, 2020

Editor: Peter Turnbaugh

Reviewer(s): Disclosure of reviewer identity is with reference to reviewer comments included in decision letter(s). The following individuals involved in review of your submission have agreed to reveal their identity: Robert jenq (Reviewer #1)

Transaction Report:

DOI: <https://doi.org/10.1128/mSystems.00857-19>

January 14, 2020

Dr. Alfonso Benítez-Páez
Institute of Agrochemistry and Food Technology Institute, Spanish National Research Council
(IATA-CSIC)
Microbial Ecology, Nutrition & Health Research Unit
C/ Catedratic Agustin Escardino, 7
Paterna-Valencia 46980
Spain

Re: mSystems00857-19 (Depletion of Blautia species in the microbiota of obese children relates to intestinal inflammation and metabolic phenotype worsening)

Dear Dr. Alfonso Benítez-Páez:

Apologies for the delay over the holidays. While the reviewers raised multiple concerns, I hope these can be addressed through revising the text and/or potentially adding additional experimental data and computational validation.

Below you will find the comments of the reviewers.

To submit your modified manuscript, log onto the eJP submission site at <https://msystems.msubmit.net/cgi-bin/main.plex>. If you cannot remember your password, click the "Can't remember your password?" link and follow the instructions on the screen. Go to Author Tasks and click the appropriate manuscript title to begin the resubmission process. The information that you entered when you first submitted the paper will be displayed. Please update the information as necessary. Provide (1) point-by-point responses to the issues raised by the reviewers as file type "Response to Reviewers," not in your cover letter, and (2) a PDF file that indicates the changes from the original submission (by highlighting or underlining the changes) as file type "Marked Up Manuscript - For Review Only."

Please return the manuscript within 60 days; if you cannot complete the modification within this time period, please contact me. If you do not wish to modify the manuscript and prefer to submit it to another journal, please notify me of your decision immediately so that the manuscript may be formally withdrawn from consideration by mSystems.

To avoid unnecessary delay in publication should your modified manuscript be accepted, it is important that all elements you upload meet the technical requirements for production. I strongly recommend that you check your digital images using the Rapid Inspector tool at <http://rapidinspector.cadmus.com/RapidInspector/zmw/>.

Corresponding authors may join or renew ASM membership to obtain discounts on publication fees.

Need to upgrade your membership level? Please contact Customer Service at Service@asmusa.org.

Sincerely,

Peter Turnbaugh

Editor, mSystems

Journals Department
Reviewer comments:

Reviewer #1 (Comments for the Author):

The authors present a study of the relationship between intestinal commensal bacteria and obesity/insulin resistance in children. This is a relatively small study (15-20 per group), but strengths include an evaluation of fecal cytokines, as well as in vitro assessment using PBMCs and cytokine readouts for potential causality of the correlations they've identified. I have the following questions and suggestions to improve their manuscript.

Is the IL4/IFN γ or IL4/TNF ratio a meaningful metric with a track record? The authors should provide the raw data values for individual cytokines, in addition to the ratios that they currently provide. Did they not measure IL10? Also, did they measure IL4 or IL10 in the fecal samples?

Plan is in place to publicly deposit sequencing data.

Can the authors discuss how the OTU associated with obesity+IR is also a Lachnospiraceae - how to reconcile with their other findings? How did it classify using the NCBI database? Are there possible genetic, metabolic, or other functional differences from other Lachno relatives that are associated with the opposite phenotype?

For the potentially beneficial *Blautia* species, are they essentially absent, or are they present at a low level? The answer to this question could guide a future interventional strategy - should the *Blautia* species be simply introduced, or rather should the diet of the patients be modified to expand endogenous *Blautia*?

Correlations are a bit tricky to interpret by p value, since the slope can be significantly non-zero when driven by just a few samples. For the most interesting correlations (best rho values), please provide a XY scatterplots.

Reviewer #2 (Comments for the Author):

Benitez-Paez et al. present an amplicon sequencing-based analysis of a small cohort of lean, obese, and obese/insulin resistant children. They find significant differences in microbial community structure, including a depletion for multiple *Blautia* species in obese children relative to lean controls. Finally, they test the impact of two *Blautia* strains on PBMCs, although the results are mixed.

The major strength of this paper is that it addresses a clinically relevant topic (childhood obesity) and a major gap in knowledge regarding the structure of the gut microbiota in teens/pre-teens. Another potential strength is the inclusion of a permutation-based R package, although additional validation is necessary. However, multiple major issues limited my enthusiasm for publication at this time.

Major issues:

1. The abstract and introduction should be re-written to better match the descriptive nature of this study. As is, the authors emphasize the need to make causal links between the gut microbiome and obesity, especially in children, but there is limited data in this study to address this knowledge gap. Instead, I would recommend emphasizing the need to better understand the microbiome in childhood and its associations with metabolic disease (which is better covered by the current data). The conclusion also needs to be revised to avoid overstating the results and their translational implications.
2. Lack of validation of the "permubiome" approach. This algorithm could potentially be of interest to other groups and I applaud the authors for making it available on github. However, it's unclear what tests have been done to validate this approach or how the results compare to more established statistical analysis (LefSe, DEseq, etc).
3. BLASTN is not typically used for assigning species-level taxonomy. The RDP Classifier or other established approaches would be preferable (see <https://benjjneb.github.io/dada2/assign.html>). If the authors prefer BLASTN, more information is needed about the protocol used for taxonomic assignment and what controls were included to assess the rate of false positives/negatives.
4. Lack of experimental data. The PBMC data is a nice start, but as is doesn't really move the story much beyond association. I'm also unclear how to interpret the observed effects since all of the groups decreased the "anti-inflammatory" marker relative to PBS controls, including LPS. Statistics are provided relative to *B. vulgatus* not PBS, which could lead the reader to make the incorrect conclusion about the impact of the bacterial products. No statistics are provided for LPS relative to *B. vulgatus* or LPS relative to the *Blautia* strains. I also found the overlapping bar graphs to be unnecessarily confusing, which bars are being compared and how do these values compare? It would be much simpler to just graph IL-4, TNFa, and INFg separately.

Minor points:

Line 56: typo - "overweigh"

Line 61: This is a vast oversimplification and seems to conflict with the goal of this study to identify bacteria that impact energy balance.

Line 77: I don't follow how these contradictory results can be resolved with the current study design.

Line 128: BLAST is capitalized.

Line 452: typo - "aknowledgements"

Line 619: I'm not sure how the multiple testing correction was applied. Also, there's a typo in "Benhamini".

Figure 1 - the PERMANOVA values should go in the text not the figure.

Figure 3 - it's unclear what corrections were done for multiple hypotheses or which comparisons are statistically significant, if any.

Reviewer #1 concerns:

The authors present a study of the relationship between intestinal commensal bacteria and obesity/insulin resistance in children. This is a relatively small study (15-20 per group), but strengths include an evaluation of fecal cytokines, as well as in vitro assessment using PBMCs and cytokine readouts for potential causality of the correlations they've identified. I have the following questions and suggestions to improve their manuscript.

R/ We appreciated the reviewer's comments highlighting the strengths of our study. We must say that this kind of clinical studies on obese children/teens represents a serious difficulty sometimes because of hesitations and unwillingness of patients themselves and/or their legal tutors. Moreover, insulin resistance is rarely measured directly in obese children or adolescents, and this is not assumed as a routinary practice in all clinical centres, which makes difficult estimate precisely the prevalence of this condition and to recruit the appropriate number of patients.

Is the IL4/IFN γ or IL4/TNF ratio a meaningful metric with a track record? The authors should provide the raw data values for individual cytokines, in addition to the ratios that they currently provide. Did they not measure IL10? Also, did they measure IL4 or IL10 in the fecal samples?

R/ IL4 is a type-2 cytokine produced by Th2 cells as IL10. Although IL10 is often used as an anti-inflammatory cytokine marker that regulates cytokine production by Th1 type cells, which in turn modulate the macrophage activation, IL4 is also known to control and contribute to IL10 production (doi: 10.1038/s41598-017-11803-y, doi: 10.14202/vetworld.2019.496-503). Therefore, IL4 acts upstream IL10 in the signaling pathway regulating the cell-mediated immune response. We considered that measuring IL4 production by PBMCs is at least as informative as IL10 to determine the anti-inflammatory response activated by the bacterial isolates tested. It could also be better since IL10 production is a downstream effect of IL4 and could be more susceptible to changes through the action of other immune mediators.

We decided to present data in terms of IL4/IFN- γ and IL4/TNF- α ratios since they have been regularly used as an indicator of the balance between anti- and pro-inflammatory response resulting from the exposure to different stimuli in immune assays in both humans and other mammals (doi: 10.1159/000371766, doi: 10.1101/502864, doi: 10.1128/JVI.78.8.4011-4019.2004, doi: 10.1016/j.jns.2017.08.1819, doi: 10.1093/infdis/jiw440). It is usually the case that an inflammatory stimulus also activates an anti-inflammatory response to counteract the inflammation. Therefore, the estimation of the ratios of anti- to pro-inflammatory markers are useful to quantify the direction of the response. We have briefly explained this aspect in the methodology used (lines 451-454). The reviewer#2 also raised concerns regarding the representation of Figure 3B; therefore, it has been updated as Figure 3C.

Plan is in place to publicly deposit sequencing data.

R/ In the section "Availability of data and material" we had already described accession numbers to all sequencing data generated in our study, including massive sequencing of 16S amplicons and partial 16S reference sequences of isolates. We encourage the reviewer#1 to verify this information stated in lines 467-476.

Can the authors discuss how the OTU associated with obesity+IR is also a Lachnospiraceae - how to reconcile with their other findings? How did it classify using the NCBI database? Are there possible

genetic, metabolic, or other functional differences from other Lachno relatives that are associated with the opposite phenotype?

R/ Lachnospiraceae family constitute a very heterogeneous group of microorganisms including several unclassified *Lachnoclostridium* and *Clostridium* species. In recent years, several members of this family have been allocated in other taxa by using genome-wide alignment approaches to compare species at nucleotide level. Therefore, the current taxonomy classification of this taxon likely includes quite different microorganisms in terms of genetic and metabolic traits that make impossible attribute to them a unique role in health and disease. As suggested by the reviewer#2, we have done further attempts to identify all OTUs with differential abundance. Using the DADA2 based approach (see lines 391-392), we have additionally identified the OTU680 as *Eubacterium hallii*. Yet we have no more precise identification for OTU27 and OTU266. However, a direct comparison of these last two sequences revealed these comprise two different and distant species (sequence identity = 92% over the entire sequences). The findings suggest these taxa included many different species which could have been associated with either lean or obese phenotypes, and identification at lower taxonomic levels is needed to draw more definitive conclusions.

We have added new information in lines 387-395 to clarify the methods used for taxonomy identification of OTUs.

For the potentially beneficial *Blautia* species, are they essentially absent, or are they present at a low level? The answer to this question could guide a future interventional strategy - should the *Blautia* species be simply introduced, or rather should the diet of the patients be modified to expand endogenous *Blautia*?

R/ Our results indicate there is lower abundance of *Blautia* species (*B. luti* and *B. wexlerae*) in obese children, but not with full depletion. Future preclinical studies would be needed to demonstrate that the administration of these bacterial species could promote a lean phenotype and demonstrate causality. However, due to the complexity of these experiments, they were out of the scope of the present study. Also, diet-based strategies that promote the growth of indigenous *Blautia* species could be a practical approach to promote a lean phenotype to be evaluated. At this regard, we have disclosed recently the impact of a wheat-bran extract rich in arabinoxylans (AXOS) on the human gut microbiota in a clinical trial using AXOS as dietary supplement. An initial assessment using the 16S amplicon sequence approach revealed that intake of ~10 g / day AXOS during four weeks boost the proportion of *Blautia* species in overweight subjects (doi: 10.1016/j.clnu.2019.01.012). In a more detailed analysis, we confirm that AXOS specifically increased the abundance of the species *B. wexlerae* and *B. obeum* species as well as of *E. hallii* (OTU680) (doi: 10.1128/mSystems.00209-19- see Fig.3). Therefore, a sustained AXOS intake could be a strategy to tackle obesity and glucose homeostasis impairment in children to be tested in the future in properly sized and prolonged studies. We have added new information in lines (268-274) to describe our further thoughts and hypothesis.

Correlations are a bit tricky to interpret by p value, since the slope can be significantly non-zero when driven by just a few samples. For the most interesting correlations (best rho values), please provide a XY scatterplots.

R/ Thanks to the reviewer for such a suggestion. We have included three scatter plots showing correlations between log-transformed microbiota and faecal cytokines data for the most negative rho values found. This information is now presented as Figure 3B and the respective legend was modified accordingly.

Reviewer #2 concerns:

The major strength of this paper is that it addresses a clinically relevant topic (childhood obesity) and a major gap in knowledge regarding the structure of the gut microbiota in teens/pre-teens. Another potential strength is the inclusion of a permutation-based R package, although additional validation is necessary. However, multiple major issues limited my enthusiasm for publication at this time.

R/ We thank the reviewer for his/her comments highlighting the main strengths of our study.

Major issues:

1. The abstract and introduction should be re-written to better match the descriptive nature of this study. As is, the authors emphasize the need to make causal links between the gut microbiome and obesity, especially in children, but there is limited data in this study to address this knowledge gap. Instead, I would recommend emphasizing the need to better understand the microbiome in childhood and its associations with metabolic disease (which is better covered by the current data). The conclusion also needs to be revised to avoid overstating the results and their translational implications.

R/ We agree with the reviewer. In consequence, we have introduced some modifications in the abstract and in the last lines of the introduction to better reflect the aims of this study (lines 77-79 and 82-84).

2. Lack of validation of the "permubiome" approach. This algorithm could potentially be of interest to other groups and I applaud the authors for making it available on github. However, it's unclear what tests have been done to validate this approach or how the results compare to more established statistical analysis (LefSe, DEseq, etc).

R/ Validation of biomarker discovery approaches by using different algorithms for making comparisons, and appropriate simulations with metadata-controlled microbiome data would represent an enormous effort that will take several months of work and is out of the scope of the present study. This particular goal is in our agenda to define explicitly its performance in terms of sensibility and specificity when compared to other reference tools, and it is planned to be completed in an independent research shortly. Nevertheless, we also agree that a comparative assessment would be useful to understand the utility of the new method we have implemented in the present study. To this end, we have added Table 3 describing a comparative analysis between Permubiome and the two most often used algorithms for biomarker discovery in microbiome research, LefSe and DESeq2. We found a consensus in the detection of 4 over the 7 OTUs detected by Permubiome. The comparative analysis of the respective scoring between detected and non-detected features indicate that Permubiome has no bias to detect more dissimilar microbiome features between classes, a priori suggesting no sensibility issues. Additional information at this regard can be found at lines 133-139 and 393-395.

3. BLASTN is not typically used for assigning species-level taxonomy. The RDP Classifier or other established approaches would be preferable (see <https://benjjneb.github.io/dada2/assign.html>). If the authors prefer BLASTN, more information is needed about the protocol used for taxonomic assignment and what controls were included to assess the rate of false positives/negatives.

R/ We have adopted the recommendation of the reviewer to use the DADA2 method for taxonomy assignment. This methodology corroborated most of the assignments presented at genus level and

further supported the identification of the two *B. wexlerae* associated OTUs (OTU299 and OTU662). Moreover, it also helped to provide species-level identification for OTU680 (previously considered Lachnospiraceae member), which was identified as a *Eubacterium hallii*-associated OTU (Figure 2). We also decided to retain the methodology originally described, but adding more details for a better understanding of it (lines 387-392).

4. Lack of experimental data. The PBMC data is a nice start, but as is doesn't really move the story much beyond association. I'm also unclear how to interpret the observed effects since all of the groups decreased the "anti-inflammatory" marker relative to PBS controls, including LPS. Statistics are provided relative to *B. vulgatus* not PBS, which could lead the reader to make the incorrect conclusion about the impact of the bacterial products. No statistics are provided for LPS relative to *B. vulgatus* or LPS relative to the *Blautia* strains. I also found the overlapping bar graphs to be unnecessarily confusing, which bars are being compared and how do these values compare? It would be much simpler to just graph IL-4, TNFa, and INFg separately.

R/ We completely agree with the reviewer that further experimental research has to be done to robustly demonstrate the role of *Blautia* species to control inflammation in the gut of obese subjects. Reviewer#1 also raised similar issues regarding the interpretation of Figure 3B. In the reply to reviewer#1, we explain the reasons for using this in vitro assay in PBMCs exposed to the different bacterial strains as a first approach to support the potential anti-inflammatory effect of *Blautia* species. The estimation of ratios of anti- to pro-inflammatory markers is generally used to identify the direction of the response of immune cells since a pro-inflammatory stimulus could also trigger an anti-inflammatory response to counter-regulate the inflammation (doi: 10.1159/000371766, doi: 10.1101/502864, doi: 10.1128/JVI.78.8.4011-4019.2004, doi: 10.1016/j.jns.2017.08.1819, doi: 10.1093/infdis/jiw440). Nonetheless, we have improved the presentation of the original data according to the reviewer's suggestion. It now appears in Figure 3C. The LPS treatment was removed from the comparative schema as it was used as a positive control of the inflammatory response in PBMCs but not for comparative purposes.

Minor points:

Line 56: typo – "overweigh"

R/ Corrected (line 56).

Line 61: This is a vast oversimplification and seems to conflict with the goal of this study to identify bacteria that impact energy balance.

R/ Modified as suggested (lines 62-64).

Line 77: I don't follow how these contradictory results can be resolved with the current study design.

R/ This paragraph has been written to describe the state-of-the-art of this particular topic without the purpose of providing a direct answer to this question with our results. Accordingly the lines 78-84 have been re-written.

Line 128: BLAST is capitalized.

R/ Corrected as suggested (line 132).

Line 452: typo – "aknowledgements"

R/ Corrected (line 483).

Line 619: I'm not sure how the multiple testing correction was applied. Also, there's a typo in "Benhamini".

R/ We have three different groups that were compared in a pairwise manner. So, we performed three comparisons at once making possible all group combinations. In such a case, multiple testing correction is mandatory to minimise the proportion of false positives based on the p-values distribution. The typo was corrected accordingly (line 688).

Figure 1 - the PERMANOVA values should go in the text not the figure.

R/ Removed as suggested.

Figure 3 - it's unclear what corrections were done for multiple hypotheses or which comparisons are statistically significant, if any.

R/ We have updated the Figure 3A to indicate with stars statistically significant correlations that were corrected for multiple comparisons ($p < 0.05$).

February 27, 2020

Dr. Alfonso Benítez-Páez
Institute of Agrochemistry and Food Technology Institute, Spanish National Research Council
(IATA-CSIC)
Microbial Ecology, Nutrition & Health Research Unit
C/ Catedratic Agustin Escardino, 7
Paterna-Valencia 46980
Spain

Re: mSystems00857-19R1 (Depletion of Blautia species in the microbiota of obese children relates to intestinal inflammation and metabolic phenotype worsening)

Dear Dr. Alfonso Benítez-Páez:

Upon initial review of this revised submission, I couldn't find the new Table 3. Please check the version of the uploaded files and resubmit when ready.

Below you will find the comments of the reviewers.

To submit your modified manuscript, log onto the eJP submission site at <https://msystems.msubmit.net/cgi-bin/main.plex>. If you cannot remember your password, click the "Can't remember your password?" link and follow the instructions on the screen. Go to Author Tasks and click the appropriate manuscript title to begin the resubmission process. The information that you entered when you first submitted the paper will be displayed. Please update the information as necessary. Provide (1) point-by-point responses to the issues raised by the reviewers as file type "Response to Reviewers," not in your cover letter, and (2) a PDF file that indicates the changes from the original submission (by highlighting or underlining the changes) as file type "Marked Up Manuscript - For Review Only."

Please return the manuscript within 60 days; if you cannot complete the modification within this time period, please contact me. If you do not wish to modify the manuscript and prefer to submit it to another journal, please notify me of your decision immediately so that the manuscript may be formally withdrawn from consideration by mSystems.

To avoid unnecessary delay in publication should your modified manuscript be accepted, it is important that all elements you upload meet the technical requirements for production. I strongly recommend that you check your digital images using the Rapid Inspector tool at <http://rapidinspector.cadmus.com/RapidInspector/zmw/>.

Corresponding authors may join or renew ASM membership to obtain discounts on publication fees. Need to upgrade your membership level? Please contact Customer Service at

Service@asmusa.org.

Sincerely,

Peter Turnbaugh

Editor, mSystems

Journals Department
Reviewer comments:

February 28, 2020

Dr. Alfonso Benítez-Páez
Institute of Agrochemistry and Food Technology Institute, Spanish National Research Council
(IATA-CSIC)
Microbial Ecology, Nutrition & Health Research Unit
C/ Catedratic Agustin Escardino, 7
Paterna-Valencia 46980
Spain

Re: mSystems00857-19R2 (Depletion of Blautia species in the microbiota of obese children relates to intestinal inflammation and metabolic phenotype worsening)

Dear Dr. Alfonso Benítez-Páez:

Thanks for fixing Table 3. Please also submit your raw data to EBI or NCBI instead of MG-RAST. The latter is unstable and can be difficult to access.

Below you will find the comments of the reviewers.

To submit your modified manuscript, log onto the eJP submission site at <https://msystems.msubmit.net/cgi-bin/main.plex>. If you cannot remember your password, click the "Can't remember your password?" link and follow the instructions on the screen. Go to Author Tasks and click the appropriate manuscript title to begin the resubmission process. The information that you entered when you first submitted the paper will be displayed. Please update the information as necessary. Provide (1) point-by-point responses to the issues raised by the reviewers as file type "Response to Reviewers," not in your cover letter, and (2) a PDF file that indicates the changes from the original submission (by highlighting or underlining the changes) as file type "Marked Up Manuscript - For Review Only."

Please return the manuscript within 60 days; if you cannot complete the modification within this time period, please contact me. If you do not wish to modify the manuscript and prefer to submit it to another journal, please notify me of your decision immediately so that the manuscript may be formally withdrawn from consideration by mSystems.

To avoid unnecessary delay in publication should your modified manuscript be accepted, it is important that all elements you upload meet the technical requirements for production. I strongly recommend that you check your digital images using the Rapid Inspector tool at <http://rapidinspector.cadmus.com/RapidInspector/zmw/>.

Corresponding authors may join or renew ASM membership to obtain discounts on publication fees. Need to upgrade your membership level? Please contact Customer Service at

Service@asmusa.org.

Sincerely,

Peter Turnbaugh

Editor, mSystems

Journals Department
Reviewer comments:

March 3, 2020

Dr. Alfonso Benítez-Páez
Institute of Agrochemistry and Food Technology Institute, Spanish National Research Council
(IATA-CSIC)
Microbial Ecology, Nutrition & Health Research Unit
C/ Catedratic Agustin Escardino, 7
Paterna-Valencia 46980
Spain

Re: mSystems00857-19R3 (Depletion of Blautia species in the microbiota of obese children relates to intestinal inflammation and metabolic phenotype worsening)

Dear Dr. Alfonso Benítez-Páez:

Lets try this one more time. I checked <https://www.ebi.ac.uk/ena/browser/view/PRJEB37005> and no records were found. Please submit one you're carefully checked that all data is publicly accessible.

Below you will find the comments of the reviewers.

To submit your modified manuscript, log onto the eJP submission site at <https://msystems.msubmit.net/cgi-bin/main.plex>. If you cannot remember your password, click the "Can't remember your password?" link and follow the instructions on the screen. Go to Author Tasks and click the appropriate manuscript title to begin the resubmission process. The information that you entered when you first submitted the paper will be displayed. Please update the information as necessary. Provide (1) point-by-point responses to the issues raised by the reviewers as file type "Response to Reviewers," not in your cover letter, and (2) a PDF file that indicates the changes from the original submission (by highlighting or underlining the changes) as file type "Marked Up Manuscript - For Review Only."

Please return the manuscript within 60 days; if you cannot complete the modification within this time period, please contact me. If you do not wish to modify the manuscript and prefer to submit it to another journal, please notify me of your decision immediately so that the manuscript may be formally withdrawn from consideration by mSystems.

To avoid unnecessary delay in publication should your modified manuscript be accepted, it is important that all elements you upload meet the technical requirements for production. I strongly recommend that you check your digital images using the Rapid Inspector tool at <http://rapidinspector.cadmus.com/RapidInspector/zmw/>.

Corresponding authors may join or renew ASM membership to obtain discounts on publication fees. Need to upgrade your membership level? Please contact Customer Service at

Service@asmusa.org.

Sincerely,

Peter Turnbaugh

Editor, mSystems

Journals Department
Reviewer comments:

March 6, 2020

Dr. Alfonso Benítez-Páez
Institute of Agrochemistry and Food Technology Institute, Spanish National Research Council
(IATA-CSIC)
Microbial Ecology, Nutrition & Health Research Unit
C/ Catedratic Agustin Escardino, 7
Paterna-Valencia 46980
Spain

Re: mSystems00857-19R4 (Depletion of Blautia species in the microbiota of obese children relates to intestinal inflammation and metabolic phenotype worsening)

Dear Dr. Alfonso Benítez-Páez:

Sorry for the hassle, we're trying to be strict about the data policy at mSystems. Congratulations on your exciting study!

Your manuscript has been accepted, and I am forwarding it to the ASM Journals Department for publication. For your reference, ASM Journals' address is given below. Before it can be scheduled for publication, your manuscript will be checked by the mSystems senior production editor, Ellie Ghatineh, to make sure that all elements meet the technical requirements for publication. She will contact you if anything needs to be revised before copyediting and production can begin. Otherwise, you will be notified when your proofs are ready to be viewed.

Sincerely,

Peter Turnbaugh
Editor, mSystems

Journals Department
Phone: 1-202-942-9338